# Identification of the Source for *Salmonella* Contamination of Carcasses in a Large Pig Slaughterhouse

**DOI:** 10.3390/pathogens10010077

**Published:** 2021-01-17

**Authors:** Hang Zeng, Geertrui Rasschaert, Lieven De Zutter, Wesley Mattheus, Koen De Reu

**Affiliations:** 1Flanders Research Institute for Agriculture, Fisheries and Food (ILVO), Brusselsesteenweg 370, B-9090 Melle, Belgium; hang.zeng@ugent.be (H.Z.); Geertrui.Rasschaert@ilvo.vlaanderen.be (G.R.); 2Department of Veterinary Public Health and Food Safety, Faculty of Veterinary Medicine, Ghent University, Salisburylaan 133, B-9820 Merelbeke, Belgium; Lieven.DeZutter@UGent.be; 3Infectious Diseases in Humans, Bacterial Diseases, Sciensano, B-1180 Brussels, Belgium; Wesley.Mattheus@sciensano.be

**Keywords:** *Salmonella*, pig carcasses, slaughterhouse, dehairing, evisceration

## Abstract

To identify the major source of *Salmonella* contamination in a pig slaughterhouse, samples were collected from the clean and unclean area and *Salmonella* isolates were further typed. Carcasses entering the clean area showed a *Salmonella* contamination rate of 96.7% in the oral cavity and 55.0% in the rectum content samples. Evisceration seemed not to be critical as the contamination rate of the carcasses was similar before (16.7%) and after (18.3%) this slaughter step. In the unclean area, a limited number of oral cavity samples were positive after bleeding, while a dramatic increase of positives was observed after dehairing. *Salmonella* was detected in up to 0.01 mL of the recycled water collected from the dehairing machine. Genotyping of *Salmonella* isolates showed that similar pulsotypes were present in the oral cavity and recycled water. Based on these observations it can be concluded that the recycled water used in the dehairing machine was the major source for the carcass contamination in this slaughterhouse.

## 1. Introduction

In the EU in 2018, salmonellosis was the second most commonly reported zoonotic disease in humans and was the most important cause of food-borne outbreaks. Moreover, pork was recognized as an important source of human salmonellosis cases [1].

*Salmonella* contamination may occur during any stage of pork production in which the slaughterhouse plays an important role [2,3,4]. At the time of slaughter, the presence of *Salmonella* in pigs is dependent on the colonization of the animals at farm level and the contamination that can occur during the transport and lairage period [5]. Pesciaroli et al., after investigating the association between the *Salmonella* numbers in pig intestines and the presence on carcasses, suggested that the reduction of *Salmonella* numbers in the intestines of slaughtered pigs would result in fewer contaminated carcasses [6]. In another study, a quantitative exposure assessment was carried out by building a modular risk model covering the pork production chain from farm to fork [7]. Those authors concluded that internal infection status and the external contamination of a pig is associated and could thus potentially contaminate the pork carcasses.

Along the pig slaughter line, several factors have been associated with *Salmonella* contamination on carcasses. Swabs of the oral cavity of carcasses after bleeding revealed a 14% *Salmonella* positivity rate [8], with these positive tests being significantly correlated with the carcass contamination during slaughter. Biasino et al. [9] also demonstrated that the oral cavity, tonsils and tongue of the pig play an important role in the contamination of the pig carcass. Scalding decreases *Salmonella* contamination on the exterior of the carcass but depends on a sufficiently high temperature of the scalding water (62 °C) [10,11]. In the next step of the slaughter process, fecal material may be spread onto the carcass surface during dehairing and may result in a rise in the level of carcass contamination [12,13]. In the clean area, the intestines and pluck set (e.g., heart, liver, lungs, tongue, etc.) are removed from the pig carcass during evisceration, which is another step where pig carcasses can become contaminated with *Salmonella*; this step is also associated in which cross-contamination [14,15,16,17,18,19,20]. Apart from the factors discussed above, steps such as polishing and carcass splitting are also critical for *Salmonella* contamination [21,22]. 

The process hygiene criterium set by European Commission Regulation 217/2014 stipulated that no more than three out of 50 sampled carcasses by swab may be contaminated with *Salmonella* [23]. If more are detected, the slaughterhouse is required to take corrective actions. According to the report of European Food Safety Authority and European Centre for Disease Prevention and Control (EFSA and ECDC), in 2018 the proportion of *Salmonella*-positive carcasses before chilling based on official controls was 2.7% in the EU [1]. *Salmonella* Typhimurium and *S.* Derby accounted for 27.3% and 61.8% of all isolates, respectively, collected from the positive carcasses [1]. Both *S.* Typhimurium and *S.* Derby were among the top five *Salmonella* serovars that caused human salmonellosis cases in the EU.

Control of *Salmonella* in pig slaughterhouses requires the identification of critical points or steps where (cross-)contamination can occur, together with modifications of the slaughter procedures/techniques to reduce the risk of contamination. Both the critical steps and the intervention plans can differ among slaughterhouses [11,24].

The aim of this study was to identify the main source of *Salmonella* contamination in a large Belgian pig slaughterhouse with a history of carcass contamination above the EU process hygiene criterium for pig carcasses. To confirm our findings, the identified source of contamination was also monitored in other slaughterhouses.

## 2. Results

### 2.1. Samples from the Clean Area

The first samples collected (first sampling event in the clean area—S1) were from 30 carcasses originating from seven farms. The most contaminated site of those carcasses was the oral cavity, with 93.3% (28/30) samples testing positive for *Salmonella*, whereas 53.3% (16/30) samples of the rectal content tested positive (Table 1). For the carcass swabs of breast and elbow combined, 16.7% (5/30) were positive before evisceration and 6.7% (2/30) after evisceration. Before evisceration, three breast and three elbow swabs were *Salmonella* positive, with one carcass contaminated at both sites. *Salmonella* Derby was the only serotype that was found in this batch of samples. In total 34 isolates were typed by Pulsed-Field Gel Electrophoresis (PFGE) and the *Salmonella* Derby isolates belonged to pulsotype F (76.5%, 26/34) and I^‡^ (23.5%, 8/34).

The second batch of samples (repeat after 6 weeks—S2) came from 30 carcasses originating from five farms. *Salmonella* was detected in the oral cavity of all of these carcasses (30/30) and in 56.7% (17/30) of the rectal content. Before evisceration 16.7% (5/30) of the carcasses were contaminated with *Salmonella* while 30% (9/30) of the carcasses were detected as positive after evisceration. After evisceration, six carcasses were contaminated with *Salmonella* on both sampled sites (breast and elbow). The 50 characterized *Salmonella* isolates were identified as *S.* Derby pulsotypes B (20.0%, 10/50), C (30.0%, 15/50), D (2.0%, 1/50) and I (22.0%, 11/50), *S.* Brandenburg (pulsotype K, 10.0%, 5/50) and *S.* Typhimurium (pulsotypes T: 14%, 7/50 and W: 2.0%, 1/50). *Salmonella* Derby was the dominant serotype (74.0%), which was found in all sample types. *Salmonella* Brandenburg (pulsotype K) was isolated only from the oral cavity and rectal content. *Salmonella* Derby was the only serotype found in the carcass swabs before evisceration; it was typed as pulsotypes B, C and I. After evisceration, *S.* Derby pulsotypes C and I and *S.* Typhimurium pulsotype T and W were isolated from the carcasses. In addition, three carcasses were contaminated with both *S.* Typhimurium pulsotype T and *S.* Derby (two pulsotype I and one pulsotype C) after evisceration. 

### 2.2. Sampling in the Unclean Area

The third batch of samples (the first sampling event in the unclean area—S3) were from 30 carcasses originating from five farms. After bleeding 16.7% (5/30) of the carcasses were *Salmonella* positive, while after dehairing all carcasses (100%) were contaminated with *Salmonella* (Table 2). More specifically, after bleeding, three swabs from the oral cavity and three swabs from the rectum were *Salmonella* positive, with one carcass testing *Salmonella* positive for both the oral cavity and the rectum. After dehairing, 28 swabs from the oral cavity and 29 swabs from the rectum were *Salmonella* positive, and from 27 carcasses, *Salmonella* was isolated from both sites. Within this batch of samples, *Salmonella* Derby pulsotype C (8.3%, 3/36) and *S.* Typhimurium pulsotypes O^‡^ (5.6%, 2/36), Z (2.8%, 1/36) were identified after bleeding. After dehairing, *S.* Derby pulsotypes C (66.7%, 24/36) and I (11.1%, 4/36) and *S.* Infantis pulsotype M (2.8%, 1/36) and *S.* Typhimurium pulsotype U (2.8%, 1/36) were identified.

In the water of the scalding tank at the beginning and the end of the sampling event, no *Salmonella* was detected (Table 3). The water temperature of the scalding tank ranged between 59.2 °C and 60.3 °C. The temperature of the water at the outlet of the three parts of the dehairing machine ranged from 28.4 °C to 38.2 °C. At the start of the sampling event, the water of the dehairing machine was positive for *S.* Derby (pulsotypes C: 26.3%, 5/19, I: 15.8%, 3/19) in up to 0.1 mL water from parts 1 and 3, and 1 mL from part 2. Approximately 1.5 h later, after sampling was completed, water samples were more contaminated with *Salmonella,* detected in 0.01 mL from parts 1 and 3, and in 0.1 mL from part 2. In addition to *S.* Derby pulsotypes C (26.3%, 5/19) and I (15.8%, 3/19), *S.* Typhimurium pulsotype R (5.3%, 1/19) and W^‡^ (5.3%, 1/19) and *S.* Infantis pulsotype M (5.3%, 1/19) were also identified in those water samples.

The fourth time samples were taken (repeat after 10 days—S4), and 30 sampled carcasses originated from two farms. *S.* Typhimurium pulsotype Q (5.0%, 1/20) was detected in 1 out of 30 (3.3%) oral cavity swabs before dehairing while after dehairing, 93.3% (28/30) oral cavity swabs were *Salmonella* positive. *Salmonella* Derby pulsotype B (10.0%, 2/20), C (5.0%, 1/20), F (5.0%, 1/20), *S.* Panama pulsotype J (5.0%, 1/20), *S.* Rissen pulsotype H (30%, 6/20) and *S.* Typhimurium O (5.0%, 1/20), W^‡^ (25.0%, 5/20), X (5.0%, 1/20) and Y (5.0%, 1/20) were identified from these samples. The temperature of the water at the outlet of the three parts of the dehairing machine ranged from 32.1 °C to 37.9 °C. Collected water samples were positive in up to 1 mL both at the beginning and the end of sampling event. In water collected at the beginning of the sampling event, *S.* Derby pulsotype C (16.7%, 2/12), *S.* Panama pulsotype J (16.7%, 2/12) and *S.* Typhimurium O (16.7%, 2/12) were detected. At the end of the sampling event, *S.* Derby pulsotypes B (16.7%, 2/12), C (8.3%, 1/12), *S.* Panama pulsotype J (8.3%, 1/12) and *S.* Typhimurium pulsotype Q (16.7%, 2/12) were isolated from the water samples.

### 2.3. Salmonella in the Water at the Outlet of the Dehairing Machine during Slaughter

The results of the first batch of water samples (S5) are shown in Table 4. Even before commencement of that day’s slaughter activities, *S.* Derby, *S.* London, *S.* Panama, and *S.* Typhimurium were detected in water sampled at the outlet of the dehairing machine (sample taken at 6:35 AM). *Salmonella* was detected in volumes of 1–10 mL. After two hours of slaughter activities, *S.* Derby pulsotype F was detected from all three parts of the dehairing machine and in higher concentrations (concentrations of 0.1–0.01 mL tested positive). After four hours of slaughter activities, the same *S.* Derby pulsotypes was found in all water samples of 0.01 mL. Temperature of sampled water ranged between 26.3 °C and 39.0 °C (Table 4).

The results of the second batch of water samples (S6) are listed in Table 5. No *Salmonella* was found in the water before slaughter commenced that day. After one hour of slaughter, *S.* Derby pulsotypes C, I, N and *S.* Typhimurium O^‡^ and S were detected in the water of all three parts of the machine in volumes of 0.1–1 mL. After four hours of processing, *S.* Derby pulsotype I, *S.* London pulsotype L and *S.* Typhimurium pulsotype V were detected in comparable volumes of water sampled.

### 2.4. Water Samples from Dehairing at Other Pig Slaughterhouses

In slaughterhouse A, *Salmonella* was not detected in the dehairing water samples before the start of the slaughter activities, but after one hour of activity *S.* Typhimurium was detected in volumes of 1 mL and after 2 h in 10 mL (Table 6). Slaughterhouse B was also *Salmonella* negative before starting slaughtering. During slaughter, *S.* Typhimurium was detected only once (after 30 min of slaughter activity) in one of both compartments of the dehairing machine. In slaughterhouse C, before the start of the slaughter activities the water was already contaminated in a 0.1 mL volume and continued to be contaminated with *S.* Typhimurium and other *Salmonella* serovars.

## 3. Discussion

In a pig slaughterhouse with a history of *Salmonella* positive carcasses, during six sampling events consecutive samples were taken to detect the origin of the *Salmonella* contamination of the carcasses. Different studies have already revealed several critical points that may cause *Salmonella* contamination of pig carcasses during slaughter, e.g., scalding and dehairing [13], evisceration [14,16], etc.

The present study identified an important source of *Salmonella* contamination of pig carcasses during slaughter. Results of two batches of samples taken six weeks apart in the clean area during evisceration did not confirm our first hypothesis that the evisceration step could be responsible for the historical contamination. The high contamination of the oral cavity (97%) and rectal content (55%) samples with *Salmonella* entering the clean area indicated high numbers of positive carcasses coming from the unclean area. Other studies reported *Salmonella* contamination rates for the oral cavity of 14.0% after bleeding [8] and 54.3% before evisceration [17], while the contamination rate for the rectal content at evisceration was 13.3% [8] and 30.9% [17], for the colon content 18.8% [25] and for the cecal content 33.8% [16], 31.4% [26] and 36.5% [27]. Moreover, carcasses were found to be *Salmonella* positive at the entrance to the clean area. Characterization of the *Salmonella* isolates showed that in most cases the same serotypes and genotypes were present in the oral cavity and the rectal content and on carcasses within the same day. These data indicated that the source of contamination had to be found either in lairage and/or in the unclean area of the slaughterhouse.

The results of the samples first taken in the unclean area revealed a dramatic increase in the *Salmonella* contamination rate of the oral cavity and rectal content during the scalding/dehairing step. *Salmonella* was not detected in the water from the scalding tank, thus this process step could not be considered as a source for the contamination of these two sites. Indeed, water in the scalding tank at 60 °C even may reduce the *Salmonella* contamination of pig carcasses [11]. A second batch of samples confirmed that the contamination of the oral cavity and rectal content occurred during the dehairing step. All water samples collected at the outlets of dehairing machine were *Salmonella* positive and contained up to at least 100 cells/mL. Characterization of isolates showed that serologically and genetically similar *Salmonella* strains were present in the water from the dehairing machine, in the oral cavity and in the rectum. These data indicated that the recycled water was the initial source of the contamination of the carcasses. Indeed, the mouth and the anus of dead pigs both relax, allowing the water used during the dehairing process to possibly enter both cavities. When the used water is contaminated and enters the oral and rectal cavity, it could contaminate both sites. When carcasses are hung up after the dehairing process, contaminated fluid from the oral and rectal cavities could leak out, leading to contamination of carcasses during subsequent steps such as polishing and evisceration.

Water samples collected before the start of slaughter activities proved that the water was *Salmonella* positive even before the day’s activities began. This indicates that the slaughterhouse’s standard cleaning and disinfection protocol (first batch of fresh water samples) was not able to eliminate the pathogen, whereas in the second batch of fresh water samples (after cleaning and disinfection performed by the aid of a specialized company) *Salmonella* was not detected. The number of *Salmonella*/mL recycled water also showed an increasing trend during slaughter. The water used in the dehairing machine was recycled during the entire slaughter day. According to the slaughterhouse, the recycled water was injected into the machine at 50 °C. When it exited the machine during slaughter activities, the water temperature varied between 30 °C and 40 °C. The recycled water, supplemented with potable water when needed, was stored in a buffer tank. The recycled water was pumped from this tank and heated before use. The water temperature when the water left the dehairing machine and when it was stored in the buffer tank allows the growth of *Salmonella*, and the water temperature after heating was not high enough to eliminate *Salmonella*. These temperatures allowed the *Salmonella* cells to survive in the recycled water and may have even encouraged them to multiply [28]. 

As a comparison, the possible impact of the dehairing step on *Salmonella* contamination in other slaughterhouses was evaluated by analyzing the water from the dehairing machine before slaughter activities began and again during slaughter. In two slaughterhouses (A and C) using recycled water *Salmonella* was found in the water at different times during production and in one of them (slaughterhouse C) *Salmonella* was even detected before the daily slaughter began. Slaughterhouse B used fresh potable water heated up to more than 50 °C; *Salmonella* was detected only once in the used water. This water could possibly become positive on occasion when a *Salmonella* positive carcass passes through. In almost all slaughterhouses in Belgium, recycled water is used on a daily basis in the dehairing machine. In practice in those slaughterhouses daily fresh water is added to a buffer tank before starting the production and further used and recycled during the day, if needed, supplemented with extra potable water during the day. This practice of recycling water after heating to a suitable temperature (over 50 °C) is authorised by the competent authorities in Belgium. Few slaughterhouses, as is the case for slaughterhouse B, uses constantly fresh potable water as this results in a lot of water consumption and high costs.

Based on this study, we believe that the sampling strategy, sample types and number of sampled animals per sampling event can also be useful in comparable case studies to detect possible main causes of *Salmonella* contaminations in pig slaughterhouses.

## 4. Materials and Methods

### 4.1. Pig Slaughterhouse

This study was conducted in a Belgian pig slaughterhouse with a slaughter capacity of approximately 550 pigs per hour. Slaughterhouses in Belgium with this slaughter speed are considered as big slaughterhouses.The slaughterhouse had sometimes failed to meet the process hygiene criterium for *Salmonella* on pig carcasses during recent years [23].

The overview of the slaughter process is shown schematically in Figure 1. In the unclean area, the pigs were stunned with CO_2_ in groups of eight to ten pigs. Then the pigs were hung on the slaughter line and bled manually. After bleeding, the pigs were scalded in a water bath for 7 min at 60 °C. Next, the hairs from the pig carcasses were removed in a dehairing machine consisting of three connected compartments with a separate water outlet at the bottom of each compartment. This water was recycled during the slaughter day. After singeing and polishing, the carcasses were transferred to the clean area for evisceration. The abdomen was opened, and the white and red organs were removed manually, after which the carcasses were split into two parts. After meat inspection, carcass dressing and classification, the carcasses were moved to the chilling room. The live weight of the slaughtered pigs varied between 90 and 110 kg.

### 4.2. Sampling Design

#### 4.2.1. Overview of Sampling Events

The timeline of the six sampling events (S1 to S6) in the slaughterhouse is given in Figure 2. To identify the main contamination source, four sampling events took place: twice in the clean area, and then twice in the unclean area. Each time, 30 slaughtered pigs were sampled at random with an interval of two minutes between carcasses. The collection of samples began 3–4 h after the start of the day’s slaughter in each case. The origin of each of the 30 carcasses of the pigs was noted. No information on the *Salmonella* status of the batches to which the sampled pigs belonged, was available. In addition, in a later stage, water samples were taken twice from the dehairing machine. Moreover, water samples were taken once in three other slaughterhouses A, B, C (A, B and C are codes of other different slaughterhouses) and before and during the slaughter activity.

#### 4.2.2. Sampling in the Clean Area

Our first hypothesis was that the carcasses had mostly become contaminated with *Salmonella* during evisceration. Therefore, samples were taken from the oral cavity when the carcasses entered the clean area, i.e., before evisceration (Figure 1: a). Besides, the carcass sites breast (200 cm^2^) and elbow (200 cm^2^) were sampled before (Figure 1: a) and after evisceration (Figure 1: c) on the same carcass. Before and after evisceration, the left side and the right side of the carcass were swabbed, respectively. Samples were taken with sponge swabs (Sponge-Stick, 3M™, Diegem, Belgium) premoistened with 10 mL of maximum recovery diluent (MRD) (Oxoid, Basingstoke, UK). Further, whole intestinal packages of the sampled carcasses were collected after removal of the white organs (Figure 1: b). The rectum was tied and excised for analysis. The same sampling protocol was used during both sampling events (S1 and S2).

#### 4.2.3. Sampling in the Unclean Area

We revised our hypothesis based on the combined results of both sampling events from the clean area, as this did not indicate a clear source of contamination. We then hypothesized that the *Salmonella* contamination took place during scalding/dehairing process. To test this, samples from 30 carcasses were collected in the unclean area during a sampling event (S3). The oral cavity of each carcass was sampled with a sponge swab premoistened with 10 mL MRD after bleeding (Figure 1: d) and dehairing (Figure 1: i). After bleeding and dehairing, also the rectum of the same carcasses was sampled with a cotton swab (a 4 cm diameter ball made of medical cotton handled with a sterile metal forceps) premoistened with MRD. In addition, water samples (100 mL) were collected at the beginning and end of the sampling event from the entrance and the exit of the scalding tank (Figure 1: e–f) and from the water outlet of the three compartments of the dehairing machine (Figure 1: h). At water sampling, the water temperature was also measured. Temperature of the scalding tank was measured directly, while an extra water sample from the dehairing machine was collected for immediate temperature measurement.

Based on these results, a second sampling event (S4) from the unclean area was carried out, to confirm that the contamination indeed occurred in the dehairing machine. Oral cavity swabs were taken just before (Figure 1: g) and after the dehairing (Figure 1: i) step from 30 carcasses. At the same time, water samples were also collected from the dehairing machine as described above. 

#### 4.2.4. Salmonella in the Water at the Outlet of the Dehairing Machine during Slaughter

During two following sampling events (S5 and S6), water samples were taken during slaughter to track *Salmonella* contamination of the recycled water used in the dehairing machine. First samples (S5) were taken after the slaughterhouse cleaned and disinfected the water recycling system and the dehairing machine, and later samples (S6) were taken again after a specialized company had cleaned and disinfected the recycling system and the dehairing machine. Water samples were taken from the water outlet of the three compartments. These samples were collected before slaughter began (at 7 AM) and after 1–2 and 4 h of slaughter activities. The before-slaughter samples were taken after the dehairing machine had been running for 15 min.

#### 4.2.5. Samples from Other Pig Slaughterhouses

In order to check if the identified source for *Salmonella* contamination was also present in other slaughterhouses, a similar protocol was used to collect water samples from the dehairing machine in three other pig slaughterhouses (A, B, C). Slaughterhouses A and C had a dehairing machine with a single compartment, while slaughterhouse B had a dehairing machine with two compartments. Slaughterhouses A and C recycled the water during the slaughter day, whereas slaughterhouse B used fresh potable water. Each slaughterhouse was visited once. The water outlet samples of the dehairing machine (two parts of the apparatus in slaughterhouse B) were collected 10 min before the start of the slaughter activities in the three slaughterhouses and after 0.5, 1 and 2 h during the slaughter process in slaughterhouse A; after 1, 2 and 3 h during the slaughter process in slaughterhouse B; and after 1, 2 and 3.5 h during the slaughter process in slaughterhouse C. The temperature of the water inside the dehairing machine was obtained from each slaughterhouse; this was 40–50 °C, over 50 °C and 25–30 °C in slaughterhouses A, B and C, respectively.

### 4.3. Salmonella Isolation

After transporting the samples to the laboratory under cooled conditions, analyses were started the same day (1.5–2 h after the sampling).

Twenty-five grams of rectal content were collected aseptically from the tied rectum and mixed with 225 mL Buffered Peptone Water (BPW) (Oxoid, Basingstoke, UK). To each sponge swab sample, 90 mL BPW was added, while 40 mL BPW was added to each cotton swab from the rectum. Afterwards, the samples were homogenized for 1 min in a Stomacher blender. For the water samples, 10 mL and 1 mL of the water samples were mixed with 90 mL and 9 mL BPW, respectively. Further, 1 mL of the 10^−1^ and 10^−2^ (only for water from the dehairing machine) dilutions of the water samples were mixed with 9 mL BPW.

The isolation method was based on ISO 6597-1 (ISO, 2017). Briefly, after 16–20 h of pre-enrichment in BPW at 37 ± 1 °C, 0.1 mL of the enriched broth was transferred in three spots on Modified Semi-Solid Rappaport-Vassiliadis (MSRV) (Oxoid) plates. These MSRV plates were incubated at 41.5 ± 0.5 °C for 24 and 48 h. After 24 h of incubation, a loopful of the migration area was transferred from each suspected plate to Xylose Lysine Deoxycholate Agar (XLD) (Oxoid) plates and incubated at 37 ± 1 °C for further detection and isolation. A further incubation until 48 h was applied for the negative MSRV plates and the same procedure was followed. From each XLD plate, one to two suspect colonies were purified and stored at −80 °C for further confirmation and typing.

The collected *Salmonella* isolates were confirmed at the genus level by polymerase chain reaction (PCR) using the primers described by Aabo et al. [29]. To identify isolates belonging to *S.* Typhimurium serotype, the primers described by Lin et al. [30] were used.

### 4.4. Characterization of the Salmonella Isolates

First, the repetitive element palindromic-polymerase chain reaction (rep-PCR) was applied to limit the number of non-Typhimurium *Salmonella* isolates to be serotyped [31]. From each rep-PCR cluster, at least one isolate was serotyped by the Belgian *Salmonella* reference laboratory using the Kauffman-White scheme (Sciensano, Brussels, Belgium).

Second, Pulsed-Field Gel Electrophoresis (PFGE) was applied according to the protocol of PulseNet [32]. Depending on the rep-PCR results, one up to five isolates per sample type and per cluster were selected randomly for PFGE with 50 U of *Xba*I enzyme (New England Biolab, Hitchin, UK). One *S.* Typhimurium isolate per sample type was also selected randomly for PFGE. The electrophoresis was applied by a Chef Mapper (Bio-Rad, Hercules, CA, USA). The obtained pulsotypes were analyzed with BioNumerics software V.7.6 (Applied Maths, Sint-Martens-Latem, Belgium) using the Pearson correlation with 2% optimization. Further, the pulsotypes were assigned to clusters and each cluster was given an alphabet letter (uppercase ‘A’ and lowercase ‘a’ indicate different pulsotypes). The symbol ^‡^ after the letter means that one extra band was presented in that isolate compared to the dominant pulsotype within the serotype.

The *Salmonella* isolates from the water samples collected in the slaughterhouses A, B and C were confirmed by PCR but were not further characterized.

## 5. Conclusions

The aim of this study was to find the main source of the *Salmonella* contamination in a slaughterhouse with a history of *Salmonella* positive carcasses. Based on the results, the recycled water used in the dehairing machine was identified as an important source for *Salmonella* contamination. In two other slaughterhouses that also use recycled water in the dehairing machine, water also tested positive for *Salmonella* at different times during production. These data indicated that slaughterhouses using recycled water in the dehairing machine may be at higher risk for contamination of carcasses with *Salmonella*. In order to reduce this risk, slaughterhouses should look for applicable methods to eliminate *Salmonella* from the recycled water before injecting it into the dehairing machine. Obtained results also indicate that sufficient attention must be given to cleaning and disinfection of the dehairing machine and the water recycling circuit after slaughter.

## Figures and Tables

**Figure 1 pathogens-10-00077-f001:**
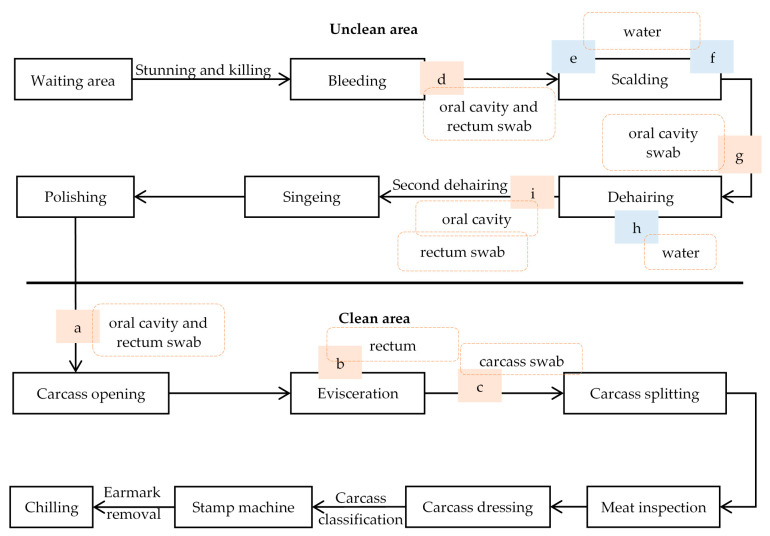
Schematic overview of the slaughter process in the pig slaughterhouse. a–i: sampling points during the different sampling events. Blue boxes refer to water samples, orange boxes refer to samples taken from the carcasses.

**Figure 2 pathogens-10-00077-f002:**
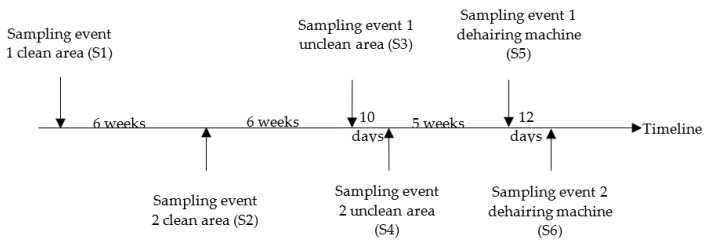
Overview of the sampling event in the pig slaughterhouse.

**Table 1 pathogens-10-00077-t001:** Presence, serotypes and pulsotypes of *Salmonella* in samples from the clean area.

Sampling Point	Before Evisceration		At Evisceration		Before Evisceration				After Evisceration			
Sample type	Oral cavity		Rectal content		Breast		Elbow		Breast		Elbow	
	No. *	S (%)—P **	No.	S (%)—P	No.	S (%)—P	No.	S (%)—P	No.	S (%)—P	No.	S (%)—P
Sampling event 1 (S1) ^†^	28	Derby (100)—F/I ^‡,&^	16	Derby (100)—F/I ^‡^	3	Derby (100)—F/I ^‡^	3	Derby (100)—F/I ^‡^	2	Derby (100)—F/I ^‡^	0	
Sampling event 2 (S2)	30	Brand ^#^ (16.7)—K Derby (83.3)—B/C/D/I	17	Brand (5.9)—K Derby (94.1)—B/I	1	Derby (100)—C	4	Derby (100)—B/C/I	7	Derby (14.3) —C Typh ^#^ (85.7)—T/W	8	Derby (75)—C/I Typh (25)—T

^†^ sampling event number; * Positive sample number, each time 30 samples were collected; ** Serotype (%: percentage of each serotype out of total positive samples) —: pulsotype; ^#^: Brand: Brandenburg; Typh: Typhimurium. ^&^: The symbol ^‡^ after the letter means that one extra band was presented in that fingerprint.

**Table 2 pathogens-10-00077-t002:** Presence, serotypes and pulsotypes of *Salmonella* in samples from the unclean area.

Sampling Point	After Bleeding				Before Dehairing		After Dehairing			
Sample type	Oral cavity		Rectal content		Oral cavity		Oral cavity		Rectum	
	No. *	S (%)—P **	No.	S (%)—P	No.	S (%)—P	No.	S (%)—P	No.	S (%)—P
Sampling event 1 (S3) ^†^	3	Derby (66.7)—C Typh ^#^ (33.3)—O ^‡,&^	3	Derby (13.3)—C Typh (66.7)—O ^‡^/Z	NS		28	Derby (100)—C/I	29	Derby (93.1)—C/I Infantis (3.4)—M Typh (3.4)—U
Sampling event 2 (S4)	NS		NS		1	Typh (100)—T4	28	Derby (17.9)—C/D/F Panama (3.6)—J Rissen (28.6)—H Typh (50)—O/W^‡^/X/Y	NS	

^†^ sampling event number; NS: Not sampled; * Positive sample number, each time 30 samples were collected; ** Serotype (%: percentage of each serotype out of total positive samples) —: pulsotype; ^#^ Typh: Typhimurium; ^&^: The symbol ^‡^ after the letter means that one extra band was presented in that fingerprint.

**Table 3 pathogens-10-00077-t003:** Presence, serotypes and pulsotypes of *Salmonella* in water samples from the scalding tank and the dehairing machine (from the unclean area).

Water Samples				Temp in ℃	Sample Analysis/mL			
					10	1	0.1	0.01
Scalding tank	Sampling event 1 (S3) ^†^	start	SP	60.3	N	N	N	NA
			EP	60.2	N	N	N	NA
		end	SP	59.7	N	N	N	NA
			EP	59.2	N	N	N	NA
Dehairing machine	Sampling event 1 (S3)	part 1	SP	38.2	Derby—C *	Derby—C	Derby—C	N
			EP	35.0	Derby—I	Derby—C	Derby—C	Derby—I
		part 2	SP	28.4	Derby—I	Derby—I	N	N
			EP	35.5	Derby—C	Infantis—M	Derby—I	N
		part 3	SP	34.8	Derby—C	Derby—C	Derby—I	N
			EP	37.5	Derby—C	Derby—C	Typh—R	Typh—W
	Sampling event 2 (S4)	part 1	SP	36.2	Derby—C	Typh ^#^—O	N	N
			EP	36.1	Derby—B	Typh—Q	N	N
		part 2	SP	32.2	Panama—J	N	N	N
			EP	32.1	Panama—J	Typh—Q	N	N
		part 3	SP	37.3	Derby—C Panama—J	Typh—O	N	N
			EP	37.9	Derby—C	Derby—B	N	N

^†^ sampling event number; SP: at the start of the sampling event; EP: at the end of the sampling event; NA: Not analyzed; N: *Salmonella* was not detected; Temp: temperature; * Serotype—Pulsotype; ^#^ Typh: Typhimurium.

**Table 4 pathogens-10-00077-t004:** Presence, serotypes and pulsotypes of *Salmonella* in water samples from the dehairing machine during the slaughter activities (first sampling event ^+^—S5 ^†^).

Sampled Time		6: 35 a.m.	9:00 a.m.	11:00 a.m.
Dehairing machine part 1	Temperature/°C smallest positive volume	26.3 °C 10 mL	39.0 °C 0.1 mL	30.4 °C 0.01 mL
	serotype—pulsotype	Panama—J	Derby—F	Derby—F
Dehairing machine part 2	Temperature/°C smallest positive volume	32.2 °C 1 mL	35.8 °C 0.1 mL	30.0 °C 0.01 mL
	serotype—pulsotype	Panama—A Typh ^#^—V	Derby—F	Derby—F
Dehairing machine part 3	Temperature/°C smallest positive volume	36.4 °C 1 mL	33.6 °C 0.01 mL	20.4 °C 0.01 mL
	serotype—pulsotype	Derby—a London—L	Derby—F	Derby—F

^†^ sampling event number; N: *Salmonella* was not detected; NA: not applicable; ^#^ Typh: Typhimurium. ^+^ Start of slaughter activities: 7 AM.

**Table 5 pathogens-10-00077-t005:** Occurrence, serotypes and pulsotypes of *Salmonella* in water samples from the dehairing machine during the slaughter activities (second sampling event ^+^—S6 ^†^).

Sampled Time		5:00 a.m.	8:00 a.m.	11:00 a.m.
Dehairing machine part 1	Temperature/°C smallest positive volume	23.2 °C (N)	36.0 °C 1 mL	39.5 °C 0.1 mL
	serotype—pulsotype	NA	Derby—C/N	Derby—I London—L Typh ^#^—V
Dehairing machine part 2	Temperature/°C smallest positive volume	29.2 °C (N)	35.8 °C 1 mL	34.7 °C 1 mL
	serotype—pulsotype	NA	Derby—C Typh—O ^‡^,^&^	London—L
Dehairing machine part 3	Temperature/°C smallest positive volume	32.5 °C (N)	39.3 °C 0.1 mL	35.8 °C 1 mL
	serotype—pulsotype	NA	Derby—C/I Typh—S	Derby—I

^†^ sampling event number; N: *Salmonella* was not detected; NA: not applicable; ^+^ Start of slaughter activities: 7 AM; ^#^ Typh: Typhimurium; ^&^: The symbol ^‡^ after the letter means that one extra band was presented in that fingerprint.

**Table 6 pathogens-10-00077-t006:** Occurrence of *Salmonella* in water samples from the dehairing machine during the slaughter activities in three other slaughterhouses A, B and C.

Slaughterhouse		Sampling Time	Before Slaughter	After 0.5 h	After 1 h	After 2 h
A	Dehairing machine	lowest positive volume PCR results	(N) NA	(N) NA	1 mL Typh.	10 mL Typh.
				After 1 h	After 2 h	After 3–4 h
B	Part 1	lowest positive volume PCR results	(N) NA	1 mL Typh.	(N) NA	(N) NA
	Part 2	lowest positive volume PCR results	(N) NA	(N) NA	(N) NA	(N) NA
C	Dehairing machine	smallest positive volume PCR results	0.1 mL Typh ^#^/*S.* spp.	0.1 mL Typh	0.1 mL Typh	0.1 mL Typh/*S.* spp.

N: *Salmonella* was not detected; NA: not applicable; ^#^ Typh: Typhimurium.

## Data Availability

Data sharing not applicable.

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
