# Peer review of "Identification of the Source for Salmonella Contamination of Carcasses in a Large Pig Slaughterhouse"

_pathogens, 2021, doi:10.3390/pathogens10010077_

Round 1

Reviewer 1 Report

This article by Hang Zeng describes the identification process of the source for Salmonella contamination of carcasses in pig Slaughterhouse.

This manuscript deserves revisions.

"et al." must be written in italic. The number below twelve must be written in full letters.

In the author's list, did the last author's name disappear?

Table 1: Authors have to prefer to indicate percentage and not an absolute number for each serotype as positive sample deeply varies between sampling event and timing.

Give readers more comparison data for the size of the slaughterhouse (little?medium?big?biggest ?)

It is very interesting that the water from water baths are recycled in some slaughtering houses. Could the authors discuss this procedure? It is clear that it sounds very risky.

More details have to be given for the choice of the sampling design? Why 6 sampling events? Why 30 pigs? Why this delay between sampling and the start of the day? What volume of water was collected ? How much time between sampling and laboratory analyses? Why have the authors decided to select only 1 to 2 suspect colonies? How was the Salmonellae been typed by the Belgian Reference LAboratory (They have to be acknowledged or included in the authors' list).

The sampling procedure has to be described once at the beginning of part 4.2 in order to not be redundant.

Why have Salmonella isolates from the water samples been confirmed by PCR but not characterized?

Reviewer 2 Report

The manuscript is well written. I have only minor recommended changes to help the authors clarify a few points in their manuscript. 

Minor Comments:

Lines 54-57: It would be helpful to further define this. For example, is this a carcass wash? Per 25 g sample?

Lines 121-122: I recommend stating: “with Salmonella detected in 0.01 mL from parts 1 and 3, and in 0.1 mL from part 2.”

Table 5: change “38,5” to “38.5”

Line 173: I recommend stating “Salmonella was not detected” instead of “tested negative.” Also in lines 176, 207, 227 and in the table footnotes.

Line 175: add a space between 10 and mL; check throughout

Figure 1: please add a description of the color coding used in your figure to the figure legend. For example, why are some sample sites marked with a blue box while others appear orange?

Round 2

Reviewer 1 Report

This article has completed most of my previous asked revision. 

Few remains and the manuscript could be accepted if fullfilled.

Point 5 : This dimension is important to be written in the manuscript, for non-specialist readers.

Point 6 : The fact that slaughterhouses investigated in this study recycled their water is very interesting and deserve to be discussed in detailed. Is this procedure frequent in other slaughterhouses? Is this procedure fully accepted by hygienist controller (or only tolerated..)?

Point 8 and 9 ; I completely agree with the comments the authors have done. My comments were also done in order to fully understand the extrapolability of the procedure? If something similar occurred in an other slaughterhouse, do the authors recommend to take this sample number? more, less? ... give details.

Point 12 : add your accepted delay to the manuscript.

Author Response

Point 1: Round 1_Point 5: This dimension is important to be written in the manuscript, for non-specialist readers.

Response 1: Following sentence was added in lines 263-264 with track changes and highlighted in green: “Slaughterhouses in Belgium with this slaughter speed are considered as big slaughterhouses.”

Point 2: Round 1_Point 6: The fact that slaughterhouses investigated in this study recycled their water is very interesting and deserve to be discussed in detailed. Is this procedure frequent in other slaughterhouses? Is this procedure fully accepted by hygienist controller (or only tolerated…)?

Response 2: The follow information about this was added in lines 249-256 with track changes and highlighted in green: “In almost all slaughterhouses in Belgium, recycled water on a daily basis is used in the dehairing machine. In practice in those slaughterhouses daily fresh water is added to a buffer tank before starting the production and further used and recycled during the day, if needed supplemented with some extra potable water during the day. This practice of re-cycling water after heating to a suitable temperature (over 50°C) is authorised by the competent authorities in Belgium. Few slaughterhouses, as is the case for slaughterhouse B, uses constantly fresh potable water as this results in a lot of water consumption and high costs.”

Point 3: Round 1_Point 8 and 9; I completely agree with the comments the authors have done. My comments were also done in order to fully understand the extrapolability of the procedure? If something similar occurred in an other slaughterhouse, do the authors recommend to take this sample number? more, less? ... give details.

Response 3: The follow information about this was added in line 257-259 with track changes and highlighted in green: “We believe that the sampling strategy, sample types and number of sampled animals per sampling event can also be useful in comparable case studies to detect possible main causes of Salmonella contaminations in pig slaughterhouses.”

Point 4: Round 1_Point 12: add your accepted delay to the manuscript.

Response 4: The time was added in line 355 with track changes and highlighted in green.